# Multi-modal Self-supervised Pre-training for Regulatory Genome Across Cell Types

## Abstract

In the genome biology research, regulatory genome modeling is an important topic for many regulatory downstream tasks, such as promoter classification, transaction factor binding sites prediction. The core problem is to model how regulatory elements interact with each other and its variability across different cell types. However, current deep learning methods often focus on modeling genome sequences of a fixed set of cell types and do not account for the interaction between multiple regulatory elements, making them only perform well on the cell types in the training set and lack the generalizability required in biological applications. In this work, we propose a simple yet effective approach for pre-training genome data in a multi-modal and self-supervised manner, which we call **GeneBERT**. Specifically, we simultaneously take the 1d sequence of genome data and a 2d matrix of (transcription factors × regions) as the input, where three pre-training tasks are proposed to improve the robustness and generalizability of our model. We pre-train our model on the ATAC-seq dataset with 17 million genome sequences. We evaluate our GeneBERT on regulatory downstream tasks across different cell types, including promoter classification, transaction factor binding sites prediction, disease risk estimation, and splicing sites prediction. Extensive experiments demonstrate the effectiveness of multi-modal and self-supervised pre-training for large-scale regulatory genomics data.

## 1 Introduction

In the gene biology research, many important regulatory problems including promoter (Li et al., 2006) classification and transaction factor binding sites (Stewart et al., 2012) prediction requires a well-designed approach for modeling regulatory genome sequences. As a multi-cellular organism, the human body is formed by different cell types, with each has its own gene expression patterns (Schena et al., 1995; Ross et al., 2000). Therefore, modeling regulatory genome, especially across different cell types, is crucial for both the understanding of this fundamental biological process and the development of RNA-level therapeutic intervention of a vast among of diseases. The main challenge is how to model the interaction between regulatory elements and its variability across different cell types.

In recent years, some works (Chen et al., 2016; Kelley et al., 2016; Romero et al., 2017; Qiu et al., 2018; Torada et al., 2019; Chen et al., 2020a; Avsec et al., 2021; Ji et al., 2021) have been proposed to apply deep learning architectures on modeling the genome data. For example, Enformer (Avsec et al., 2021) combines dilated CNN and transformer architecture as well as multi-head output to predict gene expression and epigenomic marks, and gain some performance improvements against traditional methods. Most of these models are based on a supervised learning paradigm, which limits their abilities to learn the general interaction representations between different regulatory elements. As a result, each model is only useful for some specific downstream tasks. More recently, DNABERT (Ji et al., 2021) is introduced to formulate the whole DNA sequence as a sentence of nucleotide k-mers and utilize BERT to model the sequence generatively. In this way, the interactions between different regulatory elements could be well captured, like the language modeling approach in natural language processing (NLP) to capture the co-occur information between different words, and could be employed for modeling several related downstream tasks. However, DNABERT only performs well on the cell types in the training set, and generalizes poorly to unseen cell types. This

is mainly because the whole DNA sequence is common for different cells, and pre-training on such data cannot well reflect various interaction patterns across different cell types.

Integration of genome data modalities across different cell types could help to build a more holistic model of gene expression regulation and benefit downstream applications such as mutation impact evaluation and disease risk prediction, as well as promote our understanding of the cell-type-specific regulatory programs and various development processes and disease etiology. Inspired by this idea, in this work, we present a simple yet effective method called GeneBERT, for pre-training large-scale genome data in a multi-modal and self-supervised manner. Specifically, we simultaneously take the 1D modality (*i.e.* sequence) and a 2D modality (*i.e.* regulatory region) of genome data as the input, where three pre-training tasks are proposed to improve the robustness and generalizability of our model. 1) masked genome modeling: we randomly mask some parts of the input k-mers with a special token (i.e., [MASK]), and the model is trained to predict the masked k-mers. 2) next genome segment prediction: we train the model using the embedding [CLS] to classify whether a pair of given sequences are two consecutive sequences in a cell. 3) sequence-region matching: a sequence-region matching mechanism is proposed to capture the multi-modal alignment between sequence and regulatory region of genome data.

We pre-train the GeneBERT on the ATAC-seq dataset (Domcke et al., 2020) with 17 million gene sequences. Furthermore, we conduct extensive experiments to evaluate our GeneBERT on various downstream tasks, including promoter prediction, transaction factor binding sites prediction, gene mutation localization, and personalized diseases prediction. Comprehensive ablation studies demonstrate the effectiveness of multi-modal self-supervised pre-training for large-scale genome data across different cell types.

The main contributions of this work lie in the proposal of a simple yet effective method named GeneBERT, for large-scale genome data pre-training in a multi-modal and self-supervised manner. To the best of our knowledge, we are the first to incorporate different genome data modalities across various cell types into the pre-training for large-scale genome data, to tackle the regulatory genome modeling problem. Except for meaningful biological improvements, our model makes an important contribution to the machine learning community, by introducing a novel multi-modality construction. Different from existing multi-modal learning tasks, such as visual question answering, image caption, and image-text retrieval, the 'visual' modality in our work is constructed based on the regulatory property of the 'language' sequential units. This idea brings some inspiration to building a new 'visual' modality, based on text matching, part-of-speech tagging, and named entity recognition etc., for the study of NLP.

## 2 RELATED WORK

**Uni-modal language pre-training.** Self-supervised pre-training models such as GPT (Radford et al., 2018), BERT (Devlin et al., 2018), RoBERTa (Liu et al., 2019), and ERNIE (Sun et al., 2019) have led to dramatic improvement on a variety of natural language processing tasks in the past few years, significantly surpassing the traditional context-independent language model such as Word2Vec. Obviously, BERT (Devlin et al., 2018) uses the masked language model (MLM) and next sentence prediction (NSP) for pre-training, which represent the dynamic vector of words according to the context. RoBERTa (Liu et al., 2019) uses dynamic MLM and discards NSP to train the model with a long time. ERNIE (Sun et al., 2019) masks entities and phrases to learn more context relations. These self-supervised pre-training models are capable of performing natural language processing tasks more effectively by improving MLM.

**Multi-modal pre-training.** Multi-modal pre-training has recently addressed researchers' attention for learning meaningful representations. Typically, Previous methods (Radford et al., 2021; Huo et al., 2021) learn visual representations from text paired with images in unsupervised, self-supervised, weakly supervised, and supervised ways. Since language and vision can share a similar semantic meaning, CLIP (Radford et al., 2021) is a commonly-used neural network trained on a variety of (image, text) pairs for learning transferable visual representations from natural language supervision. WenLan (Huo et al., 2021) applies a cross-modal contrastive learning framework called BriVL for image-text pre-training. However, in this work, we leverage the multi-modal self-supervised pre-training on the genome data to improve the robustness and generalizability of pre-trained models used for data-scarce scenarios.

**Genome data pre-training.** Transformer models have been recently established to better understand the genotype-phenotype relationships (Avsec et al., 2021; Ji et al., 2021). DNABERT used the human genome to pre-train a BERT-based model, trying to decipher the regulatory code related to gene expression (Ji et al., 2021). In order to adapt the DNA scenario, sequences were split into 5 to 510 base-pair long and tokenized to 3- to 6-mer representations. The model was then pre-trained using those k-mers by predicting around 15% randomly masked regions of the sequence. After the pre-training, the model was fine-tuned on three downstream tasks related to gene regulation: prediction of promoters, transcription factor binding sites (TFBSs), and splice sites. Furthermore, by analyzing the attention maps, DNABERT could visualize the important regions contributing to the model decision, which improved the interpretability of the model.

## 3  METHOD: GENEBERT

In this section, we propose a simple yet effective approach for pre-training genome data in a multi-modal and self-supervised manner, as illustrated in Figure 1. Specifically, we introduce two main objectives for sequence pre-training, including Masked Genome Modeling (MGM) and Next Genome-Segment Prediction (NGSP). Next, we extract features of the region for region pre-training. Finally, we present the Sequence-Region Matching mechanism to explicitly align features of the sequence and region in the input genome data in a contrastive self-supervised manner. Therefore, our GeneBERT consists of three main components: sequence pre-training, region pre-training, and sequence-region matching.

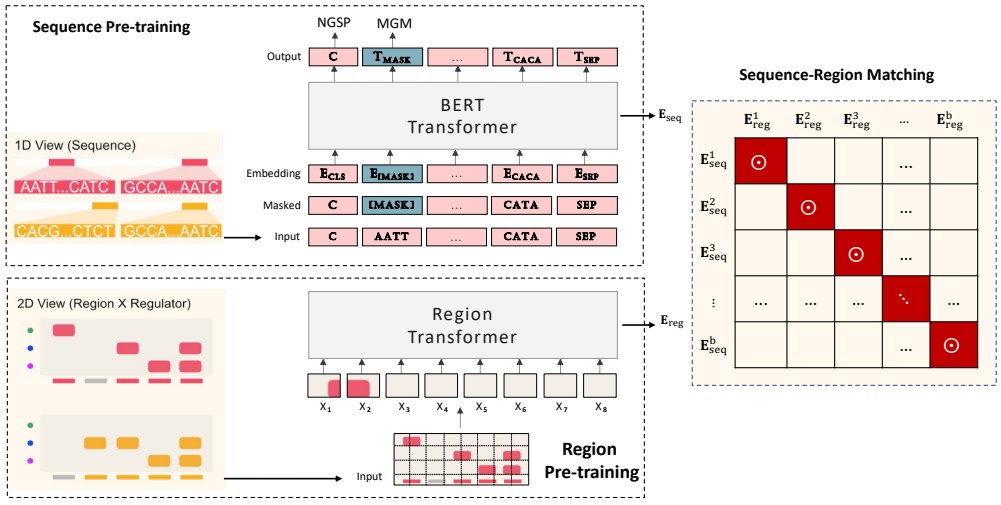

Figure 1: The overall framework of our proposed GeneBERT model.

### 3.1  INPUT DATA

We use potential regulatory elements mapped with ATAC-seq which measures the chromatin accessibility across the genome. The regions with open accessibility could be bound by transcription factors, which together with regulatory elements and RNA polymerase determines the expression level. In one cell type, the list of all regulatory elements can be viewed as a set of non-overlapped sub-regions across the genome. Each region or regulatory element can be viewed as a string like "AATTCCT..." consist of 4 elements: [A, T, C, G]. The binding site of different transcription factors has different sequence patterns, some of which have been determined using experimental methods and have been modeled using probabilistic models like position weight matrix (PWM). Using PWM, one can get the predicted affinity of a sequence segment to a given transcription factor. Each regulatory region can then be associated with a 'transcription factor binding vector" which contains the *in silico* prediction of their affinity to a fixed number of transcription factors. By scanning through a regulatory element sequence with PWM of multiple transcription factors, we can get a transcription factor binding affinity vector for that regulatory element.

When the collection of transcription factor binding models is comprehensive and precise enough, we can get a good representation of regulatory potential for a given sequence. In different cell types, the accessibility of regulatory elements is different. Some regulatory elements are shared across different cell types while some are more specific to one cell type. As a result, in a local region of the genome, the accessible transcription factor binding sites are determined by the combination of accessible regulatory regions. This combinatorial interaction between regulatory regions and transcription factors varies across different cell types such that precise control of expression in different cell types is possible. To capture the combinatorial interaction and the implicit cell-type specific information, we then group 10 consecutive regulatory elements into one train sample by concatenating their string (1D sequence) and stacking their binding vector into a 2D region, as shown in Figure 1.

### 3.2 SEQUENCE PRE-TRAINING

Similar to DNABERT, we take k-mer tokens as the sequence units. The k-mer refers to a sequence with length k, *i.e.*, for a sequence AGTCAG, the 3-mers are {AGT, GTC, TCA, CAG}, and the 4-mers are {AGTC, GTCA, TCAG}. Therefore for sequence pre-training, we input three types of embeddings: 1) a k-mer embedding $\mathbf{x}^k$ for each k-mer in a sequence; 2) a segment embedding $\mathbf{x}^s$ indicating which part of the sequence the k-mer is from; 3) a position embedding $\mathbf{x}^p$ for the position of the k-mer in the sequence. Then we sum up all three embeddings to obtain a contextual representation $\mathbf{e}^n, n \in \{1, 2, ..., N\}$, where $N$ denotes the number of k-mers in the sequence. After being fed into a BERT-based transformer $f_{\text{seq}}(\cdot)$, those contextual embeddings become $\mathbf{E}_{\text{seq}}$, i.e. the obtained representation for [CLS]. Inspired by BERT (Devlin et al., 2018), we adopt two objectives including Masked Genome Modeling (MGM) $\mathcal{L}_{mgm}$ and Next Genome Segment Prediction (NGSP) $\mathcal{L}_{ngsp}$. For $\mathcal{L}_{mgm}$, we randomly mask some parts of the input k-mers with a special token (i.e., [MASK]), and the model is trained to predict the masked kmer. The objective is formulated as:

$$\mathcal{L}_{mgm} = \max\ P(K_{mask}|K) \tag{1}$$

where $K$ and $K_{mask}$ denote the unmasked and masked k-mers, respectively. As for $\mathcal{L}_{ngsp}$, we train the model using the embedding [CLS] to classify whether a pair of given sequences are consecutive in a cell. The loss function of $\mathcal{L}_{ngsp}$ is calcualted as:

$$\mathcal{L}_{ngsp} = \texttt{CrossEntropy}(P([\text{CLS}]), y) \tag{2}$$

where $P([\text{CLS}])$ and $y$ represent the predicted probability and target for classification, respectively.

### 3.3 REGION PRE-TRAINING

For the region features in the pre-training, we consider a strong backbone (*i.e.* Swin (Liu et al., 2021)) transformer [1] as the region encoder $f_{\text{reg}}(\cdot)$ to extract representations $\mathbf{E}_{\text{reg}}$. Specifically, we apply the Swin transformer pre-trained on ImageNet to the region input directly to generate $\mathbf{E}_{\text{reg}}$. During the pre-training, we unfreeze the parameters of the Swin transformer and update them for learning better regional representations. In the pre-training setting, each region input corresponds to each sequence such that we can capture the multi-modal alignment between sequence and region of genome data.

### 3.4 SEQUENCE-REGION MATCHING

In order to learn the alignments between sequence and region of genome data, we propose a sequence-region matching mechanism based on sequence embeddings $\mathbf{E}_{\text{seq}}$ and region embeddings $\mathbf{E}_{\text{reg}}$. Firstly, we calculate the similarity (*i.e.* inner product) between each pair of 'linguistic' embeddings $\mathbf{E}_{\text{seq}}^i$ and 'visual' embeddings $\mathbf{E}_{\text{reg}}^i$ in a batch of size $b$, where $i \in 1, 2, ..., b$. Then, those similarities are jointly learned for alignments between the whole sequence and each region in a contrastive learning manner, where we maximize the similarity of the sequential and regional embeddings of the $b$ correct pairs in the batch while minimizing the similarity of the embeddings of the $b(b-1)$ false pairs. Specifically, an info-NCE like loss defined as follows is applied over these similarities scores for optimization:

$$\mathcal{L}_{srm} = -\log \frac{\sum_{i=1}^b \mathbf{E}_{\text{seq}}^i \cdot \mathbf{E}_{\text{reg}}^i}{\sum_{i=1}^b \sum_{j=1}^b \mathbf{E}_{\text{seq}}^i \cdot \mathbf{E}_{\text{reg}}^j} \tag{3}$$

---

[1]For other transformer strcutures, we explore them in the ablation study.

---

**Algorithm 1** GeneBERT multi-modal pre-training algorithm

---

    **Input:** Sequence transformer $f_{\text{seq}}(\cdot)$, Region transformer $f_{\text{reg}}(\cdot)$
1: 1D-data: k-mer embedding $\mathbf{x}^k$, segment embedding $\mathbf{x}^s$, position embedding $\mathbf{x}^p$
2: 2D-data: Region matrix $\mathbf{X}$
3: Initialize the parameters $f_{\text{seq}}(\cdot), f_{\text{reg}}(\cdot)$ with pre-trained weights
4: **for** each iteration step **do**
5:     # Sequence Pre-training
6:     Feed $\mathbf{x}^k, \mathbf{x}^s, \mathbf{x}^p$ into $f_{\text{seq}}(\cdot)$ to generate $\mathbf{E}_{\text{seq}}$
7:     calculate $\mathcal{L}_{mgm}$ and $\mathcal{L}_{ngsp}$ in Eq. 1 and 2
8:     # Region Pre-training
9:     Feed 2D region $\mathbf{X}$ into $f_{\text{reg}}(\cdot)$ to generate $\mathbf{E}_{\text{reg}}$
10:     # Sequence-Region Matching
11:     Apply embeddings $\mathbf{E}_{\text{seq}}, \mathbf{E}_{\text{reg}}$ to calculate the loss in Eq. 3
12:     Compute the total loss in Eq. 4
13:     Update the parameters of $f_{\text{seq}}(\cdot), f_{\text{reg}}(\cdot)$
    **Output:** $f_{\text{seq}}(\cdot), f_{\text{reg}}(\cdot)$

---

In this way, alignments between the whole sequence and each region are learned via our GeneBERT in the pre-training process.

Combining the above three losses together, we obtain the overall objective function for GeneBERT:

$$\mathcal{L} = \mathcal{L}_{mgm} + \mathcal{L}_{ngsp} + \lambda \cdot \mathcal{L}_{srm} \qquad (4)$$

where we vary $\lambda$ in [0.01, 1] to perform the parameter study for $\lambda$, and observe that the performance of our model is stable when $\lambda$ lies in the region of [0.5, 1]. Therefore in our experiments, we directly set $\lambda = 0.5$. The overall algorithm is summarized in Algorithm 1.

## 4 EXPERIMENTS

### 4.1 PRE-TRAINING DATA & SETTINGS

For pre-training data, we process public human fetal cerebrum single-cell chromatin accessibility data in the Descartes database (Domcke et al., 2020) to generate pseudo-bulk accessibility tracks for each cell type (Seurat cell clustering provided by the original paper). Specifically, we take the provided "Peak Count Sparse Matrices" and summed up columns (cells) according to cell type definition, producing a regions × cell-types matrix. Then we binarize the matrix and use only non-zero entries (accessible regions) for each cell type. The corresponding sequence for each region is then retrieved from the HG19 human reference genome, a single representation of multiple genomes. While the motif scanning for each region is either retrieved from the Descartes database or scanned following the same approach using JASPAR 2018 (Khan et al., 2018) vertebrate transcription factor binding site motifs. In total, we use 17 cell types and the union of all accessibility track includes 1,000,029 accessible regions across the genome, covering 504,657,456 base pairs. For the 1D modality, we group 10 consecutive accessible regions into one sample, which corresponds to a (10 × number of TFs) matrix for the 2D modality. Following previous works (Ji et al., 2021), we pre-train the model for 120k steps with a warm-up learning rate of 4e-4 and batch size of 2000. 15% of k-mers in each sequence are masked in the first 100k steps and 20% for the last 20k steps.

### 4.2 DOWNSTREAM TASKS & RESULTS

**Promoter Classification.** Promoters are the elements responsible for regulating the initial transcription of the gene, which is located near the transcription start site (TSS). As the promoters play an important role in gene regulation, using machine learning methods to predict promoter sites accurately is one of the most popular problems in bioinformatics. Here we first used the promoter core dataset from (Ji et al., 2021), which are the 70bp sequences centered around TSS. The promoter core is the key part of the promoter flanking region which is sufficient to direct accurate initiation of transcription (Oubounyt et al., 2019). Here we fine-tune our GeneBERT model to predict the promoter core sequences and compared them with DNABERT. We report the experimental results

in Table 1. For a fair comparison with DNABERT, we reproduce the results of DNABERT by fine-tuning the pre-trained models provided by the authors. During the reproduction process, we observe that it is hard to get the same results reported in the original paper as their model is very sensitive to the hyper-parameter settings. From the comparison results, we can see that our model can predict promoter core more precisely with improved performance when compared to DNABERT. This is due to the inclusion of regulatory genome information in our 2D pre-training data, which guides the pre-training process to focus on those binding site motifs and their interactions with each other.

Table 1: Comparison results on promoter classification.

| Method | Precision | Recall | AUC |
|---|---|---|---|
| DNABERT | 0.675 | 0.637 | 0.693 |
| GeneBERT (ours) | **0.805** | **0.803** | **0.894** |

Table 2: Comparison results on Transcription Factor Binding Sites classification.

| Transcription Factor & Cell Line | Method | Precision | Recall | AUC |
|---|---|---|---|---|
| CTCF_A549_CTCF_UT-A | DNABERT | 0.250 | 0.500 | 0.501 |
| | GeneBERT (ours) | **0.908** | **0.899** | **0.983** |
| CTCF_A549_CTCF_UW | DNABERT | 0.250 | 0.500 | 0.542 |
| | GeneBERT (ours) | **0.925** | **0.921** | **0.983** |
| CTCF_AG04449_CTCF_UW | DNABERT | 0.250 | 0.500 | 0.523 |
| | GeneBERT (ours) | **0.907** | **0.894** | **0.983** |
| CTCF_AG04450_CTCF_UW | DNABERT | 0.250 | 0.500 | 0.501 |
| | GeneBERT (ours) | **0.929** | **0.925** | **0.987** |
| CTCF_AG09309_CTCF_UW | DNABERT | 0.250 | 0.500 | 0.545 |
| | GeneBERT (ours) | **0.931** | **0.927** | **0.987** |
| CTCF_AG09319_CTCF_UW | DNABERT | 0.250 | 0.500 | 0.529 |
| | GeneBERT (ours) | **0.924** | **0.919** | **0.983** |
| CTCF_AG10803_CTCF_UW | DNABERT | 0.250 | 0.500 | 0.535 |
| | GeneBERT (ours) | **0.934** | **0.932** | **0.981** |
| CTCF_AoAF_CTCF_UW | DNABERT | 0.250 | 0.500 | 0.531 |
| | GeneBERT (ours) | **0.917** | **0.913** | **0.982** |
| CTCF_BE(2)-C_CTCF_UW | DNABERT | 0.250 | 0.500 | 0.540 |
| | GeneBERT (ours) | **0.937** | **0.935** | **0.989** |

**Transcription Factor Binding Sites (TFBS) Classification.** Predicting TFBS is an important step in studying gene regulation. Sequencing technologies like ChIP-seq can provide information on the in vivo binding sequences, which improve the identification of gene regulatory regions. There are several previous studies that tried to predict TFBSs using traditional machine learning (Hong & Yip, 2020) and deep learning methods (Alipanahi et al., 2015). By incorporating the multi-modal pre-training, the prediction of TFBSs can be further improved. Although we utilize the motif information during the region pre-training, we do not provide any matching information to the model, which avoids leaking information about the actual motif of a specific TF. Here we fine-tune our model for predicting TFBSs from the ChIP-seq data, using 497 TF ChIP-seq uniform peak profiles from ENCODE Consortium (Consortium et al., 2012). Note that the data in this task is different from that reported in DNABERT, since they did not provide the data for this task. We reproduce the results of DNABERT on this TFBS data with the same setting as our GeneBERT. We take the peak sequences of each TF as the positive set and generated a corresponding negative set by randomly shuffling the nucleotides in each positive sequence while preserving dinucleotide frequencies. Again, our model can accurately predict the CTCF binding sites in all tested cell lines.

**Disease Risks Estimation.** GeneBERT could provide more interpretations of complex genetic diseases. On the one hand, while the disease status and genomic mutations were available, by integrating the 2D data, the relationships among regulatory regions of genes could be captured, which allowed us to estimate the disease risk more accurately. As shown in Table 3, GeneBERT can precisely predict Hirschsprung Disease (HSCR), which is known as a genetic disorder with complex patterns of inheritance. Particularly, our GeneBERT outperforms the baseline by a large margin in terms of all metrics (Precision, Recall, AUC), which further demonstrates the effectiveness of our approach in the treatment target sites and proceeded to the medical experimental validation.

Table 3: Comparison results on disease risks estimation.

| Data | Method | Precision | Recall | AUC |
|------|--------|-----------|--------|-----|
| HSCR-RET | DNABERT | 0.265 | 0.500 | 0.500 |
| | GeneBERT (ours) | **0.770** | **0.519** | **0.562** |
| HSCR-RET-Long | DNABERT | 0.252 | 0.500 | 0.462 |
| | GeneBERT (ours) | **0.768** | **0.513** | **0.541** |

**RNA-Splicing Sites Prediction.** RNA splicing is important for post-transcription processing to remove introns from pre-mRNA sequences and generate mature mRNA for protein translation. Previously, Dilated CNN models have been used to predict splice junction across the genome and evaluate the impact of genomics variants on splicing sites (Jaganathan et al., 2019). In particular, for each nucleotide in a given sequence for splicing site prediction, we follow the previous approach and include a context sequence around the nucleotide, which could potentially capture the sequence specificity features of RNA-binding proteins and splicing machinery. Since open chromatin regions and splicing sites do not always overlap with each other, among all 548,000 splicing sites in the GTEx pre-mRNA transcripts data, our pre-training sequence only fully covers the entire (in the 256nt context setting) sequence of 72,500 sites. The experimental results are reported in Table 4. In total, 26.7% of nucleotides in context and splicing site sequence are included in the open chromatin region that we use for pre-training. Following the same training/testing split scheme and classification metric (Top-k Accuracy and PR-AUC) as in the SpliceAI study (Jaganathan et al., 2019), we achieve comparable or better results in different context settings without an extremely long context sequence. These results clearly demonstrate the generalizability of our GeneBERT pre-trained representations. By integrating sequence binding features of RNA binding proteins, we are able to further expand our model to enable cell-type specific splicing junction prediction in the future.

Table 4: Comparison results on Splicing datasets.

| Data | Method | Top-k Accuracy | PR-AUC |
|------|--------|----------------|--------|
| SpliceAI-80nt | dilated CNN | 0.57 | 0.60 |
| | GeneBERT (ours) | **0.83** | **0.89** |
| SpliceAI-256nt | dilated CNN | - | - |
| | GeneBERT (ours) | **0.93** | **0.95** |
| SpliceAI-400nt | dilated CNN | 0.90 | 0.95 |
| | GeneBERT (ours) | **0.95** | **0.98** |
| SpliceAI-2k | dilated CNN | 0.93 | 0.97 |
| | GeneBERT (ours) | **0.97** | **0.99** |

# 5 ABLATION STUDY

In this section, we perform extensive ablation studies on the effect of each module on the final performance of our GeneBERT, the effectiveness of modeling different cell types, the effect of 2D modality on pre-training, and the effect of 2D transformers on the final performance of our approach. Unless specified, we conduct all ablation studies on the promoter classification task and report the mean value of all results with 5 random seeds.

**Effect of each loss.** In order to explore the effect of each loss on the final performance of our GeneBERT, we ablate each loss in our method and report the comparison results of promoter classification in Table 5. As can be seen, pre-training without three proposed losses achieves the worst results in terms of all metrics. This means that pre-training without any objectives causes embeddings with insignificant genome information. Adding $\mathcal{L}_{mgm}$ first boosts the performance by 0.659, 0.625, 0.672 for precision, recall, and AUC. By combining $\mathcal{L}_{mgm}$ and $\mathcal{L}_{ngsp}$, we achieve the performance gain of 0.028, 0.024, 0.032, which validates the rationality of $\mathcal{L}_{mgm}$ and $\mathcal{L}_{ngsp}$ on modeling the meaningful embeddings of genome sequences. In the meanwhile, adding $\mathcal{L}_{srm}$ to the final objective further improves the results by 0.103, 0.142, and 0.169, in terms of all metrics. This further demonstrates the effectiveness of our sequence-region matching loss in learning the alignments between sequence and region of genome data. In this case, the sequence-region matching loss is helpful for multi-modal self-supervised pre-training on large-scale genome data, which boosts the performance of downstream tasks.

Table 5: Ablation study on each loss.

| $\mathcal{L}_{mgm}$ | $\mathcal{L}_{ngsp}$ | $\mathcal{L}_{srm}$ | Precision | Recall | AUC |
|---|---|---|---|---|---|
| ✗ | ✗ | ✗ | 0.016 | 0.012 | 0.021 |
| ✓ | ✗ | ✗ | 0.675 | 0.637 | 0.693 |
| ✓ | ✓ | ✗ | 0.703 | 0.661 | 0.725 |
| ✓ | ✓ | ✓ | **0.805** | **0.803** | **0.894** |

Table 6: Exploration study on different cell/tissue types.

| Protein | Cell type | Tissue type | Precision | Recall | AUC |
|---|---|---|---|---|---|
| AG04449 | Fibroblast | Skin | 0.907 | 0.894 | 0.983 |
| AG04450 | Fibroblast | Lung | 0.929 | 0.925 | 0.987 |
| AG09309 | Fibroblast | Skin | 0.931 | 0.927 | 0.987 |
| AG09319 | Fibroblast | Gingival | 0.924 | 0.919 | 0.983 |
| AG10803 | Fibroblast | Skin | 0.934 | 0.932 | 0.981 |
| AoAF | Fibroblast | Heart | 0.917 | 0.913 | 0.982 |
| BE(2)-C | Neuroblast | Brain | **0.937** | **0.935** | **0.989** |

**Generalization to different cell/tissue types.** To validate the generalizability of our approach to various cell or tissue types, we evaluate our GeneBERT on Transcription Factor Binding Sites (TFBS) classification. Specifically, we choose different proteins from various cells or tissues in the human body, such as skin, lung, heart, brain, etc. The experimental results of TFBS classification on different cell/tissue types are reported in Table 6. In terms of modeling different cell/tissue types, DNABERT (Ji et al., 2021) performs poorly on the generalizability, as we reported in Table 2. Enformer Avsec et al. (2021) only performs well on the gene expression prediction for the single cell. From the results, we can observe that our GeneBERT achieves superior performance on various cell types and tissue types in terms of all metrics, including precision, recall, and AUC. This indeed shows the advantage of our GeneBERT on learning generalizable embeddings for different cell types and tissue types in the human body. Particularly, we achieve the best performance on the brain tissue and neuroblast cell, which might be caused by the pre-training data from the human brain.

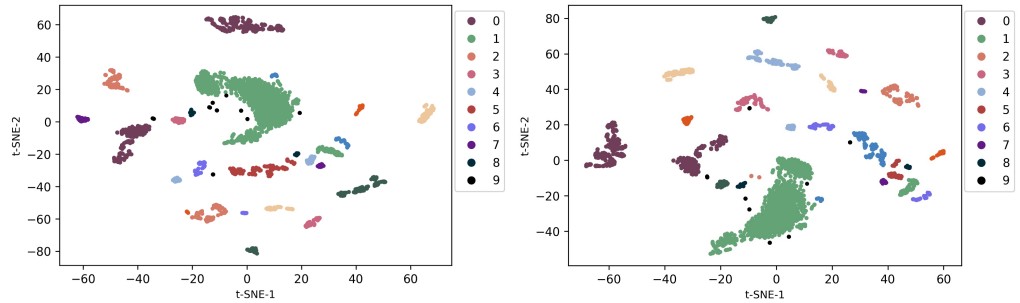

Figure 2: Visualization results of representations pre-trained by the uni-modal (**left**) and multi-modal (**right**) manner. Different colors denotes the various cell types.

**Effect of 2D modality on pre-training.** In order to further evaluate how the 2D modality data affect the quality of pre-training on large-scale genome data, we visualize the pre-trained representations pre-trained by the uni-modal and multi-modal manner in Figure 2. Typically, we project the embeddings pre-trained from BERT onto 2 dimensions using tSNE tools, and we select randomly 10 cell types in the pre-training data for visualization. We can observe that our GeneBERT pre-trained representations form more separated clusters that are distributed more uniformly on the space in terms of all cell types when compared to the pre-trained representations generated by the uni-modal model. Furthermore, the representations pre-trained by our multi-modal self-supervised method have much smaller distances inside each cluster of cell type. This further demonstrates the effectiveness of our 2D modality in improving the quality of pre-trained representations.

**Effect of different transformer strcutures.** In order to figure out why we choose the strong Swin transformer to take the 2D modality data as input, we explore the effect of different 2D transformers on the final performance of our GeneBERT. Specifically, we apply four commonly-used backbones

(ViT (Dosovitskiy et al., 2021), T2T-ViT (Yuan et al., 2021), DeiT (Touvron et al., 2020), Swin (Liu et al., 2021)) as the 2D transformer to extract the region features, and report the comparison results in Table 7. As can be seen, our GeneBERT with Swin-B-384 backbone achieves the best performance in terms of all metrics (precision, recall, AUC). In the meanwhile, with the increase of input size, the performance of our GeneBERT improves, which demonstrates the effectiveness of features extraction of the 2D modality data in learning discriminative representations of large-scale genome data. As the 2D transformer gets powerful, our GeneBERT achieves better results in terms of all metrics on the promoter classification task. This further validates the importance of the 2D transformer in multi-modal self-supervised learning. In this case, we choose the Swin transformer as the 2D backbone since we achieve the best results with the Swin transformer.

Table 7: Exploration study on 2D transformers with different input sizes.

| 2D Transformer | Input Size | Precision | Recall | AUC |
|---|---|---|---|---|
| ViT-B-384 | 384 | 0.758 | 0.755 | 0.853 |
| ViT-L-384 | 384 | 0.752 | 0.753 | 0.851 |
| T2T-ViT-14 | 224 | 0.805 | 0.803 | 0.892 |
| T2T-ViT-24 | 224 | 0.808 | 0.806 | 0.896 |
| DeiT-S-224 | 224 | 0.788 | 0.795 | 0.873 |
| DeiT-B-224 | 224 | 0.806 | 0.805 | 0.895 |
| DeiT-B-384 | 384 | 0.809 | 0.808 | 0.897 |
| Swin-T-224 | 224 | 0.805 | 0.803 | 0.894 |
| Swin-S-224 | 224 | 0.813 | 0.811 | 0.905 |
| Swin-B-224 | 224 | 0.822 | 0.823 | 0.915 |
| Swin-B-384 | 384 | **0.824** | **0.826** | **0.919** |

**Effect of batch size.** Batch size is always a significant hyper-parameter in the contrastive learning literature. To explore how the batch size affects the final performance of our multi-modal self-supervised pre-training, we vary the batch size from 16, 32, 64, 128, 256, and 512. The experimental results are reported in Table 8. We can observe that with the increase of the batch size, our GeneBERT achieves better performance due to the effectiveness of more negative samples in the sequence-region matching loss. This demonstrates the importance of a larger batch size and more negative samples on large-scale genome data pre-training, which complies with the conclusion of vision tasks in (Chen et al., 2020b;c; He et al., 2020; Chen et al., 2020d).

Table 8: Ablation study on the paramters, batch size and $\lambda$.

| Batch Size | Precision | Recall | AUC | $\lambda$ | Precision | Recall | AUC |
|---|---|---|---|---|---|---|---|
| 16 | 0.642 | 0.651 | 0.733 | 0.01 | 0.633 | 0.621 | 0.683 |
| 32 | 0.751 | 0.752 | 0.853 | 0.05 | 0.703 | 0.702 | 0.772 |
| 64 | 0.783 | 0.792 | 0.871 | 0.1 | 0.781 | 0.762 | 0.851 |
| 128 | 0.805 | 0.803 | 0.894 | 0.5 | **0.805** | **0.803** | **0.894** |
| 256 | 0.812 | 0.813 | 0.906 | 1.0 | 0.802 | 0.801 | 0.892 |
| 512 | **0.815** | **0.817** | **0.908** | 5.0 | 0.785 | 0.782 | 0.852 |

**Effect of $\lambda$.** Furthermore, we explore the effect of $\lambda$ on the final performance of our GeneBERT. Specifically, we vary $\lambda$ from 0.01, 0.05, 0.1, 0.5, 1.0 and 5.0, and report the comparison results in Table 8. Our GeneBERT achieves stable performance when $\lambda$ is set to 0.5 and 1. With the decrease of $\lambda$, the performance of our approach deteriorates in terms of all metrics, which validates the importance of our sequence-region loss in our GeneBERT. To receive the stable performance of multi-modal self-supervised pre-training, we set $\lambda = 0.5$ in all experiments.

# 6 CONCLUSION

In this work, we present the GeneBERT, a multi-modal self-supervised framework for large-scale genome data pre-training. Specifically, we leverage sequence pre-training, region pre-training, and sequence-region matching together to improve the generalizability of our model across different cell types for regulatory genome modeling. Extensive experiments on four main regulatory downstream tasks demonstrate the effectiveness of our GeneBERT. Besides, comprehensive ablation studies have been conducted to show the important role of multi-modal modeling.

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
