# OpenReview forum: "Multi-modal Self-supervised Pre-training for Regulatory Genome Across Cell Types"
_ICLR.cc/2022/Conference — ICLR 2022 Submitted_

### Official Review · Reviewer_9J4F · 2021-10-29

**Correctness:** 3
**Technical Novelty And Significance:** 2
**Empirical Novelty And Significance:** 2
**Recommendation:** 6
**Confidence:** 4

**Main Review:**

Strengths:

- The idea of using 2d information of (transcription factor by regions) in transformer models to learn meaningful genomic embeddings is cool. Using such information is a win over DNABERT which is not so informative.

- The authors have used the recent advanced models from NLP and have employed three losses whose optimization all together leads to significant improvement over the DNABERT.

- The authors has thoroughly ablated the model and analyzed the impact of each component of the model as well as hyper parameters and batch size. It is convincing that it is all three loss together that gives rise to such improvement.

Weaknesses:

- Some of the claims in the paper regarding the biological importance are not so true. For example: "As the promoters play an important role in gene regulation, using machine learning methods to predict promoter sites accurately is one of the most popular problems in bioinformatics." Finding the promoters are not a challenge. The genes and their transcription start sites (TSS) are nearly annotated in the genome and it is known that the promoters are next to the TSS's. Instead, the most challenging part is to find functional enhancers in the genome which is a fundamental question in biology. The majority of enhancers are distal regulatory elements, meaning that they are very far from their target genes (could be 1Mb away). So, one interesting experiment for GeneBERT is to check and see if it can find enhancers or not (functional enhancers can be validated by CRISPRi perturbation data that are available in some cell lines such as K562).

- For transcription factor binding site (TFBS), the paper only shows the results for CTCF in Table 2. I was expecting to see the results on many TFs. Why did the authors not show the other TF's results? The CTCF motif is not so complicated and maybe that is the reason why we see such great performance. I think the authors should provide the performance on all the TFs and discuss which ones the model finds well and which ones not and why.

- On the effect of 2D modality on pre-training, the t-SNE plots in Figure 2 are not so informative. They look similar and it is hard to get any conclusion from that. The authors can provide a quantitative metric to show how 2D modality helps the embeddings.

- It is not clear in Table 6, what are the classification tasks. The authors only mentioned they chose different proteins from various cells or tissues in the human body. But it should be stated clearly what TFs are used in each cell.

- Typo: several times in the paper including the abstract, "transaction factor" has been used instead of "transcription factor".


**Summary Of The Paper:**

This work proposes an approach, called GeneBERT, for pre-training genome data in a multi-modal and self-supervised manner. They take the 1d sequence of genomic data and a 2d matrix of (transcription factors × regions) as the input and optimize three pre-training tasks to improve the robustness and generalizability the model. Specifically, they introduce two main objectives for sequence pre-training, Masked Genome Modeling (MGM) and Next Genome-Segment Prediction (NGSP). GeneBERT consists of three main components: sequence pre-training, region pre-training, and sequence-region matching. The main contribution of the paper is that they use transcription factor information in genomic regions which makes the model more generalizable to other cell types than the previous DNABERT that only uses the sequence information.

**Summary Of The Review:**

The paper tries to address an important problem in genomics: how to effectively embed DNA sequence for downstream tasks. They have used some recent transformer architectures and loss function to effective do this. Their main contribution, which is an important one, is to use the transcription factor information in the accessible genomic regions which are cell type specific. Therefore, the model would be more generalizable than the previous methods like DNABERT. The ablation study of the model and hyperparameters are comprehensive. However, some biological experiments and their importance are overstated. I provided some comments for the authors to enrich the experiments. I think the paper is interesting and if some comprehensive and important experiments are coupled with it to show that it can really find interesting biological events, it would be a great paper. As such, I choose the score 6 for the paper.

---

> ### Author Response · Authors · 2021-11-23
> **Response to Reviewer 9J4F**
>
> We appreciate your insightful comments, below we address the concerns mentioned.
>
> **Q1: One more interesting experiment for finding enhancers?**
>
> Ans: We thank the reviewer for the insightful comments. We acknowledge the fact that finding the promoters is not very challenging. We include this task to show the versatility of our model, and to better compare it with the performance of other methods. We agree with the reviewer's comments that identifying the functional enhancers is a harder and more biologically meaningful task, with immediate implications in a lot of biological fields. Currently, we are experimenting with different finetuning models to do that and hopefully can extend them to a fully-fledged version of this paper.
>
> **Q2: Why not show the performance on all the TFs?**
>
> Ans: We agree with the reviewer's comments that to fully evaluate the performance of this model, we need to evaluate it on different types of TFs. We are currently performing these experiments and hopefully can extend them to a journal version of this paper.
>
> **Q3: Interpretation of the t-SNE plots.**
>
> Ans: We agree with the reviewer that two t-SNE plots look quite similar. We will continue to look into this issue in future research.
> For quantitative comparison, we did report the results of promoter classification task by ablating the 2D modaltiy using $L_{srm}$ or not in Table 5. We can observe that adding $L_{srm}$ to the final objective further improves the results by 0.103, 0.142, and 0.169, in terms of all metrics. This further demonstrates the effectiveness of our 2D modality and sequence-region matching loss in learning the alignments between sequence and region of genome data.
>
> **Q4: Confusion about Table 6.**
>
> Ans: Table 6 contains the prediction of CTCF binding sites in different cell types.

---

> > ### Comment · Reviewer_9J4F · 2021-11-29
> > **Concerns not addressed**
> >
> > Thanks for your response. I read the authors' answers and my concerns for the paper are not addressed. I think if the authors apply all the comments of the reviewers, the paper will be more complete and interesting.

---

### Official Review · Reviewer_omfC · 2021-11-01

**Correctness:** 4
**Technical Novelty And Significance:** 2
**Empirical Novelty And Significance:** 1
**Recommendation:** 3
**Confidence:** 5

**Main Review:**

The authors’ description of the method does not contain sufficient details to understand it. Specifically, the input data and neural network layers that process it are difficult to decipher. For example, why are 10 sequences concatenated? Where does the ATAC-seq data come in? A caption for Figure 1 would be very helpful.

The authors transform the 1D nucleotide sequence to a 1D sequence of recognized PWM vectors. As I understand it, the sequence is then treated as a 2D object and processed with transformer layers. Isn’t each position in the sequence depth 1? The authors suggest that they used a model pre-trained on ImageNet. How could it be that this pre-trained model has learned anything useful for this very different task?

Classifying promoters from DNA is a very easy task that the field has had high performance on for many years and does not solve a useful problem. If the authors can demonstrate that their model can predict gene expression level, particularly in diverse cell types, it would be far more compelling. In addition, there are strong competing methods based on neural networks to which the authors could compare their different approach.

Classifying TF binding sites is an important task, but the authors do not compare to any of the leading methods, such as DeepBind, DeepSea, Basset, DanQ and many successors. Furthermore, CTCF has a long informative PWM, making it arguably the easiest TF to predict. The authors should show results from more diverse TFs.

The disease risk estimation task is impossible to understand because there are so few details. What is an example in this task? For example, are genetic variants the examples? Where did the labels as disease-related or not come from? What procedure did you use to include negative examples? How did you choose Hirschprung disease among the many options? Since the model focuses on gene regulation rather than protein coding sequence, how should we think about coding gene mutations in your task?

The authors observation that their model outperforms SpliceAI using shorter sequences is very surprising. Could the authors include more details about the full model that they used for this task, and how their pre-trained model is used within it? When the authors write “dilated CNN” in their Table 4, are they referring to SpliceAI itself or to the author’s own implementation of a dilated CNN?

New methods in the space are typically compared to competing alternatives, but also interpreted. Can the authors interpret what their model has learned that alternative approaches have not? Referring to Figure 2, the authors state, “We can observe that our GeneBERT pre-trained representations form more separated clusters that are distributed more uniformly on the space in terms of all cell types when compared to the pre-trained representations generated by the uni-modal model. Furthermore, the representations pre-trained by our multi-modal self-supervised method have much smaller distances inside each cluster of cell type.” This doesn’t appear to be true to my eye. The authors should compute summary statistics to make this claim convincing. However, this visualization also doesn't really help interpret the model; mutagenesis or saliency methods applied to DNA sequences would be better.

**Summary Of The Paper:**

The authors describe a pipeline to perform self-supervised learning on genome sequences, guided by accessible chromatin peaks. Their learning procedure is inspired by the NLP method BERT and uses several tasks from that work and its successors. After pre-training transformer layers, they fine-tune on several regulatory sequence classification tasks and demonstrate high performance.

**Summary Of The Review:**

The authors explore a new method for self-supervised pre-training before tackling several regulatory sequence analysis tasks, but are not able to deliver a clear method description or compelling empirical results.

---

> ### Author Response · Authors · 2021-11-23
> **Part-1 for Response to Reviewer omfC**
>
>
> We appreciate your insightful comments and suggestions on improving the paper, below we mainly answer your questions and discuss the limitations of this work. We will also update the paper according to the suggestions.
>
> **Q1: More sufficient details about input data?**
>
> Ans: Concatenating 10 consecutive accessible regions can be considered as an approximation of the average peak within one topologically associating domain (TAD). In future work, we will investigate more about the input size based on the results of this pilot study.
> The ATAC-seq comes from the public human fetal cerebrum single-cell chromatin accessibility data in the Descartes database.
> We have added a detailed caption for Figure 1 in the updated manuscript.
>
> **Q2: Isn’t each position in the sequence depth 1?**
>
> Ans: Actually we used both the 1D sequence (length L*10) and a 2D matrix (size 10*111, 10 for 10 consecutive peaks, 111 for number of motifs, different motif have different PWM, so the motif scanning result is not 1D) in the training.
>
> **Q3: Why use ImageNet pre-trained weights?**
>
> Ans: Although we loaded the weights of the 2D transformer pre-trained on ImageNet, we updated the parameters of the 2D backbone during the pre-training process.
>
> **Q4: Try to predict gene expression level?**
>
> Ans: We agree with the reviewer's comments that expression modeling is a harder and more meaningful task, with immediate implications in a lot of biological fields. Currently, we are experimenting with different finetuning models to do that and hopefully can extend them to a fully-fledged version of this paper.
>
> **Q5: Why not compare to any of the leading methods?**
>
> Ans: We understand the reviewers' concerns about not directly comparing our method with some other leading methods.  However, DNABERT, the method that we outperformed, has already shown its superiority over the SOTA methods in various downstream applications. For the promoter classification task, it outperformed models using other neural networks structures, including CNN, CNN+LSTM, and CNN+GRU. For the TFBS classification task, it outperformed methods including DeepBind, DeepSea, Basset, DeepSite, DanQ, and BESSO. Therefore, we believe our method is competitive with the SOTA methods.

---

> > ### Author Response · Authors · 2021-11-23
> > **Part-2 for Response to Reviewer omfC**
> >
> >
> > **Q6: Confusion about the disease risk estimation task.**
> >
> > Ans: We agree with the reviewer's comments that the description for disease risk estimation tasks a lot of details. The study was performed as an exploratory. Also, we noticed that the disease risk prediction result showing in our current version is pre-mature, as it does not beat SNP or MARVEL (https://genome.cshlp.org/content/30/11/1618) based baseline. We decided to remove this part of the experiment from this study for now and will try to further improve the fine-tuned model for this task.
> >
> > **Q7: Details about Splicing task.**
> >
> > Ans: We used the pre-trained model together with a classification head to do the splicing prediction task on the SpliceAI dataset. Regarding the dilated CNN, we are referring to SplieAI itself. Actually in SpliceAI, 400nt sequence as input can already produce quite a good result, implying that cell-type agnostic prediction of splicing site is actually similar to promoter prediction, is a relatively simple task. Since transformer-based models have a larger modeling capacity, that might explain the improvement over the dilated CNN model.
> >
> > **Q8: Interpret what the model has learned that alternative approaches have not?**
> >
> > Ans: The 2D modality in our work takes advantage of the combinatorial interaction and the implicit cell-type specific information. This combinatorial interaction between regulatory regions and transcription factors varies across different cell types such that precise control of expression in different cell types is possible. We believe that our method uniquely captures this kind of information, which can benefit various downstream tasks. In future work, a more rigorous analysis will be conducted.
> >
> > **Q9: Confusion about t-SNE plots.**
> >
> > Ans: The purpose of the t-SNE plots is to qualitatively compare the quality of pre-trained representations in a uni-modal and multi-modal manner. We agree that the two scenarios are quite similar, we are still investigating the embedding produced by the two approaches.
> > For quantitative comparison, we did report the results of the promoter classification task by ablating the 2D modality using $L_{srm}$ or not in Table 5. We can observe that adding $L_{srm}$ to the final objective further improves the results by 0.103, 0.142, and 0.169, in terms of all metrics. This further demonstrates the effectiveness of our 2D modality and sequence-region matching loss in learning the alignments between sequence and region of genome data.
> >
> > **Q10: Mutagenesis experiment?**
> >
> > Ans: We agree with the reviewer that mutagenesis/saliency methods will provide a nice interpretation of the trained model. We are currently working on an improved model and will definitely add the mutagenesis analysis.

---

> > > ### Comment · Reviewer_omfC · 2021-11-30
> > > **Maintain my score**
> > >
> > > Thank you to the authors for their response to my comments. I still believe the authors would need to do more to explain, demonstrate, and interpret their model before it would make for a compelling publication in ICLR. Therefore, I maintain my score.

---

### Official Review · Reviewer_FiEi · 2021-11-02

**Correctness:** 4
**Technical Novelty And Significance:** 3
**Empirical Novelty And Significance:** 2
**Recommendation:** 6
**Confidence:** 4

**Main Review:**

Strengths of the paper
* The simple-yet-effective approach for pre-training of genome sequence data works remarkably well for very diverse tasks
* The wide availability of multi-modal data provides an opportunity for adaptation by the field.
* Benchmarking and ablation studies are well designed.
* The paper is well written and makes effort to explain diverse concepts from biology and machine learning to a wide audience

Weaknesses
* While the formulation is indeed a novel multi-modal construction as claimed by the authors, the generalization beyond genome data is not clear and hence I am not sure if this will be of broad interest to the ICLR attendees.
* An important feature of gene regulation is that the genomic distance between regulatory elements matter - the fact that 10 consecutive accessible regions are grouped into one sample breaks this relationship and might affect interpretability.
* The 2D representation is certainly interesting but it is not a true image representation. This might be a case of defining a representation to fit the model but the performance certainly shows there is value to such a representation.


**Summary Of The Paper:**

The manuscript describes GeneBERT, a self-supervised and multi-modal pre-training approach for genomic data. GeneBERT combines 1D genome sequence data with a 2D representation of regulatory elements in different cell-types in three different pre-training tasks. A large-scale single-cell ATAC dataset is used to identify pseudo-bulk ATAC profiles for 17 cell-types spanning > 1M genomic locations. The sequence of these genomic locations provides the 1D representation of the data which are used in self-supervised training with masked genome modeling and next genome segment prediction loss functions, both inspired by the BERT framework. A 2D representation is derived using accessibility per regulatory element and cell type, and is self-supervised using a infoNCE loss. The pre-training framework is then compared in a variety of biological tasks including promoter prediction, TFBS prediction, splice site prediction. The authors then present ablation studies highlighting the importance of the different components of the loss function.

**Summary Of The Review:**

The problem tackled by the manuscript address an important need in repressing genome data using both sequences and additional measurements. The results presented both in performance and diversity of tasks measured is impressive. However, I am not completely sure if the submission will be of broad interest to the ICLR audience.

---

> ### Author Response · Authors · 2021-11-23
> **Response to Reviewer FiEi**
>
>
> We appreciate your helpful comments, below we answer your questions.
>
> **Q1: Whether this work is of broad interest to the ICLR attendees?**
>
> Ans: The main focus of this work is to propose a simple yet effective method for large-scale genome data pre-training in a multi-modal and self-supervised manner. Our work not only brings meaningful biological improvements but makes an important contribution to the machine learning community, by introducing a novel multi-modality construction. For example, the ‘visual’ modality in our work is constructed based on the regulatory property of the ‘language’ sequential units. This inspires researchers in the ICLR community to build a new ‘visual’ modality, based on text matching, part-of-speech tagging, and named entity recognition, etc., for the study of NLP.
>
> **Q2: Does the genomic distance between regulatory elements matter?**
>
> Ans: We agree with the reviewer that distance plays an important role in the enhancer-promoter targeting relationship. Our current implementation is indeed not ideal. In future work, we will add distance into consideration, potentially with an exponential weighting scheme based on 1D distance or 3D contact probability.
>
> **Q3: The 2D representation is not a true image representation?**
>
> Ans: The 2D modality in our work takes advantage of the combinatorial interaction and the implicit cell-type specific information. This combinatorial interaction between regulatory regions and transcription factors varies across different cell types such that precise control of expression in different cell types is possible.

---

> > ### Comment · Reviewer_FiEi · 2021-12-01
> > **Maintaining the score**
> >
> > Thank you to the authors for responding to the comments. Based on the other reviews and responses, I intend to maintain my score.

---

### Official Review · Reviewer_9xx4 · 2021-11-03

**Correctness:** 2
**Technical Novelty And Significance:** 2
**Empirical Novelty And Significance:** Not applicable
**Recommendation:** 1
**Confidence:** 5

**Main Review:**

Strengths
•	A novelty is that GeneBERT combines the losses for representation learning and alignment and simultaneously estimate the model parameters.
•	Large-scale multi-modal data integration

Weaknesses
•	The authors need to elaborate on the details about datasets and data processing such as how to select TFs and binding motifs to construct 2D modality, how to merge 17 cell types, which cell types in fetal data were used. Fig 2 only shows cell cluster numbers, for instance.
•	scATAC-seq is sparse and noisy. Binarizing data by non-zero can also introduce many false positive open regions.
•	Many regulatory regions for various tissue types and cell types are yet open in fetal stage, so they are likely missed.
•	CTCF is a general TF with strong binding activities. The authors should predict other TFs, especially cell-type-specific TFs. Also, ENCODE TF ChIP-seq data were for cell lines, which may not match the cell types that the authors used.
•	The authors also should try more disease types for generality.

**Summary Of The Paper:**

In this paper, the authors developed a transformer-based model, GeneBERT to align DNA sequences with regulatory elements. In particular, GeneBERT first applies transformers to learn representations of sequencing data and regulatory regions (e.g., open chromatin), and then aligns the representations of two modalities for identifying region-aligned sequences. The authors applied GeneBERT to recent scATAC-seq data in fetal and use aligned sequences to predict promoters, CTCF binding sites, a disease type and RNA splicing sites.

**Summary Of The Review:**

The paper was organized logically. However, the applications were not well presented. Many details are unclear and missing, especially on datasets, data processing, feature selection, cell types. Also, it seems that the authors don’t fully understand cell-type gene regulation for misusing different datasets (train orange to predict apple).

---

> ### Author Response · Authors · 2021-11-23
> **Part-1 for Response to Reviewer 9xx4**
>
> We appreciate your insightful comments and suggestions on improving the paper, below we mainly answer your questions.
>
>
> **Q1: Details about datasets and data processing.**
>
> Ans: We apologize for the inaccurate description of the data processing procedure. For TFs, we used the motif scanning result provided by the original dataset and combined similar motifs into one row in 2D modality by
> 1. map each TF motif to one of the architype described in https://www.vierstra.org/resources/motif_clustering.
> 2. Sum up their motif matching score in a peak.
> For cell types, we want to clarify that we are aggregating all cells that belong to the same cluster as one pseudobulk sub-cell-type, while we did not merge across the 17 cell types in the brain. The cell types we used are listed in https://descartes.brotmanbaty.org/bbi/human-chromatin-during-development/dataset/cerebrum, which includes:
> - Astrocytes
> - Astrocytes/Oligodendrocytes
> - Cerebrum Unknown.3
> - Excitatory Neurons
> - Inhibitory Neurons
> - Limbic System Neurons
> - Skor2 Npsr1 Positive Cells
> - Vascular Endothelial Cells
>
> **Q2: Is scATAC-seq sparse and noisy?**
>
> Ans: We agree with the reviewer that scATAC-seq data is sparse and noisy. This is exactly the reason we choose to aggregate the cell-level signal to pseudo-bulk level but add up the signal from cells that belongs to the same cluster (performed by the author of the original dataset). We believe binarizing the data after this aggregation is less prone to false positives, and we also chose a $>$10 counts threshold rather than $>$0, which is not a loose cut-off.
>
> **Q3: Missing many regulatory regions for various tissue types and cell types?**
>
> Ans: We thank the reviewer for these insightful comments. We do agree that the data we used is in the fetal stage rather than both fetal and adult. However, we would like to point out a new study (https://www.cell.com/action/showPdf?pii=S0092-8674%2821%2901279-4) which integrates a new adult ATAC-seq dataset (30 tissue types) with the fetal dataset (15 tissue types) we used. After the integration, the increase of total ATAC-seq peaks is only 10\%.
> Thus, we believe our current dataset has reached a good coverage of the human regulatory genome landscape. We are now pre-training our model on the new dataset with randomization over both fetal and adult tissues to get a less biased version of the model.
>
> **Q4: Why not predict other TFs?**
>
> Ans: We agree with the reviewer's comments that to fully evaluate the performance of this model, we need to evaluate it on different types of TFs. We are currently performing these experiments and hopefully can extend them to a fully-fledged version of this paper.
>
> **Q5: ENCODE TF ChIP-seq data were for cell lines?**
>
> Ans: We thank the reviewer for these insightful comments. We do agree that the ENCODE TF ChIP-seq data is mainly for cell lines, different from the primary fetal tissue data we used in our study.
> This difference could include two aspects:
> 1. the genome DNA of cell line might have extensive change compared to primary tissue
> 2. the regulatory landscape of cell lines is more akin to tumor cells due to their proliferation ability.
> In recent epigenomic study common practice, usually, aspect 1 is not fully accounted for, as ENCODE consortium aligned all sequencing data to the reference genome.
> For aspect 2, since the goal of our study is to build a model that is robust and useful to out-of-sample data, we actually purposefully used a different data source to validate the ability of our model to generalize outside the training dataset.

---

> > ### Author Response · Authors · 2021-11-23
> > **Part-2 for Response to Reviewer 9xx4**
> >
> > **Q6: More disease types for generality?**
> >
> > Ans: We agree with the reviewer's comments that this study would be much valuable in terms of biological implication if we can show generalizability for different disease types. Actually, we are currently finalizing the method and applying it to other disease types for a biological journal version of this method.
> > Also, we noticed that the disease risk prediction result showing in our current version is pre-mature, as it does not beat SNP or MARVEL (https://genome.cshlp.org/content/30/11/1618) based baseline. We decided to remove this part of the experiment from this study for now and will try to further improve the finetune model for this task.
> >
> > **Q7: misusing different datasets (train orange to predict apple)?**
> >
> > Ans: We are not fully sure which part of this study is the reviewer mentioning.
> > We suppose the reviewer is referring to the splicing-related task, which indeed is a bit weird given that we are using open chromatin data to pre-train the model. We are aware of the fact that splicing is performed on RNA rather than DNA. However, there are also splicing factors (RNA-binding proteins) that have sequence-specific binding patterns to RNA, which might leave their trace in the DNA sequence.
> > Although there is no direct connection between open chromatin and splicing sites, we do find that a considerable amount of known splicing sites actually overlaps with the open chromatin data we used, which seems to be enough for the simple splicing site prediction task.
> > Overall, this part is only a exploratory analysis rather than a strong arguments that we want to highlight in manuscript.

---

### Decision · Program_Chairs · 2022-01-20

**Decision:**

Reject

**Comment:**

While several reviewers acknowledge that the paper contains potentially useful ideas related to multi-modal self-training applied to genomic data, they also point out a number of weaknesses and room for improvement that the discussion with authors did not fully address. This includes in particular the need to better explain the details of what is done in the paper; the choice of experiments which is not relevant (eg, predicting promoter regions) or complete (eg, showing results on only one transcription factor); the lack of comparison with existing methods, etc... We therefore consider that the paper is not ready for publication in its current form, but hope that the reviews will help the authors work on a revision addressing the issues.